# RR Myelo POINT: A Retrospective Single-Center Study Assessing the Role of Radiotherapy in the Management of Multiple Myeloma and Possible Interactions with Concurrent Systemic Treatment

**DOI:** 10.3390/cancers14092273

**Published:** 2022-05-02

**Authors:** Andrea Emanuele Guerini, Alessandra Tucci, Filippo Alongi, Eneida Mataj, Angelo Belotti, Paolo Borghetti, Luca Triggiani, Ludovica Pegurri, Sara Pedretti, Marco Bonù, Davide Tomasini, Jessica Imbrescia, Alessandra Donofrio, Giorgio Facheris, Navdeep Singh, Giulia Volpi, Cesare Tomasi, Stefano Maria Magrini, Luigi Spiazzi, Michela Buglione

**Affiliations:** 1Department of Radiation Oncology, University of Brescia and Spedali Civili Hospital, Piazzale Spedali Civili 1, 25123 Brescia, Italy; a.e.guerini@gmail.com (A.E.G.); e.mataj@unibs.it (E.M.); luca.triggiani@unibs.it (L.T.); ludovicapegurri@libero.it (L.P.); sara.pedretti@asst-spedalicivili.it (S.P.); marcolorenzo89.mlb@gmail.com (M.B.); tomad88@libero.it (D.T.); imbresciajessica@gmail.com (J.I.); alessandra.donofrio90@gmail.com (A.D.); giorgio.facheris@gmail.com (G.F.); singh.nav92@gmail.com (N.S.); g.volpi50@gmail.com (G.V.); stefano.magrini@unibs.it (S.M.M.); michela.buglione@unibs.it (M.B.); 2Department of Haematology, ASST-Spedali Civili Hospital, 25123 Brescia, Italy; alessandra.tucci@asst-spedalicivili.it (A.T.); angelo.belotti@asst-spedalicivili.it (A.B.); 3Advanced Radiation Oncology Department, Sacro Cuore Don Calabria Hospital, IRCCS Ospedale Sacro Cuore Don Calabria, 37024 Negrar Di Valpolicella, Italy; filippo.alongi@unibs.it; 4Department of Medical and Surgical Specialties, Radiological Sciences and Public Health, Section of Public Health and Human Sciences, University of Brescia, 25123 Brescia, Italy; cesare.tomasi@live.com; 5Medical Physics Department, ASST Spedali Civili Hospital, 25123 Brescia, Italy; luigi.spiazzi@unibs.it

**Keywords:** myeloma, radiotherapy, concurrent, chemotherapy, biologic therapy, immunotherapy, toxicity, pain control, multiple myeloma, disease control

## Abstract

**Simple Summary:**

Currently, few papers have been published regarding the possible interactions between radiotherapy and systemic agents for the treatment of multiple myeloma. In this paper, we retrospectively analyze the data from 312 patients (577 lesions) who received radiotherapy at our institution from 2005 to 2020, with the aim of clarifying the clinical impact of radiotherapy dose and concurrent systemic treatment (CST). The safety profile of the radiotherapy was excellent; high biologically effective doses (BEDs) and CST were associated with higher toxicity rates at the end of radiotherapy, but not after one and three months. The pain control rate was 87.4% at the end of treatment and further increased at three and six months. Radiological progression was reported only for 4.4% of the lesions at six months (based on the data available for 181 lesions) and was significantly more frequent for lesions treated without CST or BED < 15 Gy.

**Abstract:**

Background and purpose: Although chemotherapy, biological agents, and radiotherapy (RT) are cornerstones of the treatment of multiple myeloma (MM), the literature regarding the possible interactions of concurrent systemic treatment (CST) and RT is limited, and the optimal RT dose is still unclear. Materials and methods: We retrospectively analyzed the records of patients who underwent RT for MM at our institution from 1 January 2005 to 30 June 2020. The data of 312 patients and 577 lesions (treated in 411 accesses) were retrieved. Results: Most of the treated lesions involved the vertebrae (60%) or extremities (18.9%). Radiotherapy was completed in 96.6% of the accesses and, although biologically effective doses assuming an α/β ratio of 10 (BED 10) > 38 Gy and CST were significantly associated with higher rates of toxicity, the safety profile was excellent, with side effects grade ≥2 reported only for 4.1% of the accesses; CST and BED 10 had no impact on the toxicity at one and three months. Radiotherapy resulted in significant improvements in performance status and in a pain control rate of 87.4% at the end of treatment, which further increased to 96.9% at three months and remained at 94% at six months. The radiological response rate at six months (data available for 181 lesions) was 79%, with only 4.4% of lesions in progression. Progression was significantly more frequent in the lesions treated without CST or BED 10 < 15 Gy, while concurrent biological therapy resulted in significantly lower rates of progression. Conclusion: Radiotherapy resulted in optimal pain control rates and fair toxicity, regardless of BED 10 and CST; the treatments with higher BED 10 and CST (remarkably biological agents) improved the already excellent radiological disease control.

## 1. Background

Although it is considered a rare cancer, multiple myeloma (MM) is diagnosed in over 100,000 patients each year worldwide, and its prevalence is increasing [1]. Although chemotherapy and systemic agents constitute the cornerstone of its treatment, radiotherapy (RT) also has an established role in its management [2]. Osteolytic lesions are present in up to 80% of cases at diagnosis and often invade surrounding tissue, with possible spinal cord and nerve-root compression [3]. As MM is highly radiosensitive, RT allows a high rate of local response and is thus beneficial in the control of multiple complications of the disease, including pain, structural instability, and spinal cord compression [4].

In the last fifteen years, the approval of innovative systemic treatments (including bortezomib, carfilzomib, ixazomib, lenalidomide, pomalidomide, and daratumumab) has led to an impressive improvement in overall survival [5]. The advent of novel agents has also opened new perspectives for the concomitant administration of RT [6], as possible synergy has been observed in pre-clinical studies [7], but there is also concern over the possible sensitization of normal tissues [8]. Nonetheless, only a few studies, with limited populations, have been published regarding the concurrent use of RT and newly approved drugs [9,10]. This, combined with isolated case reports of severe side effects [11], might lead to the unnecessary discontinuation of systemic treatment or to depriving patients of the benefits of RT, due to fear over unexpected toxicities.

We therefore designed a retrospective analysis of our center’s experience to evaluate the effect of RT on the symptomatic and radiological control of the disease and the clinical impact of the concomitant administration of systemic therapy (CST) and RT, focusing on newly approved agents.

## 2. Methods

We retrospectively evaluated records of all the patients who underwent radiation therapy on MM lesions during the period between 1 January 2005 and 30 June 2020 at our institution. Data were retrieved from the electronic medical record systems and picture archiving and communication system used by our Institution. An access was defined as a presentation at our institution for a new radiation therapy prescription for one or more sites of disease.

For each patient, information regarding RT and eventual concurrent systemic treatment were reported, as well as information regarding number of previous systemic treatment lines, age, and sex.

Systemic treatment was defined as concurrent if the last administration was performed before the start of RT, with an interval of less than five half-lives of the drug, regardless of the line of treatment and the time from the start of treatment.

Patients were classified in three categories, according to systemic treatment administered concurrently with RT: (a) no concurrent therapy, (b) concurrent chemotherapy, (c) concurrent biologic treatment (including proteasome inhibitors, immunomodulating agents and/or monoclonal antibodies) given alone or in combination with chemotherapy.

For patients treated with combination regimens, only drugs administered with an interval of less than five half-lives from the start of RT were considered as concurrent.

Patients were also divided into three groups, according to BED_10_ (biologically effective dose assuming an α/β ratio of 10) of RT: <15 Gy, 15–38 Gy, >38 Gy.

The following outcomes were evaluated:(a)Incidence of treatment toxicity during RT and at one and three months after the end of RT and percentage of RT suspension attributable to toxicity. Toxicities were graded according to National Cancer Institute Common Terminology Criteria for Adverse Events, version 5.0.

Only toxicities with a causal relationship with RT were reported.

Our analysis included events with the following causal relationships. (a) Possible: Some evidence suggesting a causal relationship, with other factors that might have contributed. (b) Probable: Evidence to suggest a causal relationship, influence of other factors is unlikely. (c) Definite: Clear evidence to suggest a causal relationship, and other possible contributing factors can be ruled out.

Side effects with no causal relationship with RT were excluded from the analysis.

(b)Pain control at the end of RT and at one, three, and six months after RT; pain control was defined as partial, complete, or absent and possibly assessed by the patient’s self-rated pain verbal numeric scale, VNS, in which patients are asked to verbally state a number between 0 (no pain) and 10 (worst imaginable pain); data regarding reduction or variation in analgesic intake were also included, when available, for the assessment of pain control.

(c)Local radiologic in-field response at six months after the end of RT (defined according to RECIST 1.1 or PERCIST 1.0).

This protocol was developed, and the study was conducted, according to the criteria established by the 1964 Helsinki declaration and its later amendments, as well as by the ICH Good Clinical Practice, and approved by the Ethics Committee of our Hospital (number of approval NP 4595, received 1 April 2021).

Statistical analysis was performed using IBM-SPSS^®^ software ver. 26.0.1 (IBM SPSS Inc. Chicago, IL, USA). The use of Stata^®^ software ver. 16.0 (Stata Corporation, College Station, TX, USA) was also considered for comparisons or implementations of test output.

Normality of the distributions was assessed using the Kolmogorov–Smirnov test. Categorical variables were presented as frequencies or percentages and compared with the use of the chi-square test or the Fisher’s exact test, as appropriate. Continuous variables were compared with the use of Wilcoxon paired test. Associations of the crosstabs were verified using standardized adjusted residuals. A two-sided α level of 0.05 was used for all tests.

## 3. Results

The data from 312 patients (158 males and 154 females) were retrieved, with 577 lesions treated in 411 accesses (mean number of treated lesions per access 1.4, median 1). The characteristics of the patients, the diseases, and their treatment are summarized in Table 1. The median age at the start of RT was 69.8 years; the majority of the treated lesions involved the vertebrae (60%) or the extremities (18.9%). Steroid use during RT was reported for 62.9% of the lesions and 55.1% of the accesses.

The performance status, according to the Karnofsky score, significantly improved by the last day of RT (data available for 377 accesses: mean 70 before RT vs. 71.7 on the last day of RT, *p* < 0.001) with no significant impact of CST or BED_10_.

The pain reported, according to the VNS, improved from the beginning of RT to the last day of treatment (data available for 139 accesses: mean 4.8 vs. 1.7, median 5 vs. 0, *p* < 0.001) with no significant impact of BED_10_ (*p* = 0.060) or CST (*p* = 0.103).

The most common side effects were gastro-intestinal (G1 for 13.7% of accesses, G2 for 2%), esophagitis (G1 10.2%, G2 0.7%), pharyngodinia (G1 5.6%), infections (G1 4.9%, G2 1.2%), and mucositis (G1 2.4%, G2 0.2%). Only three G3 toxicity events were reported (one case of esophagitis in a patient administered concurrent chemotherapy and two infections, one in a patient undergoing RT alone, and one in a patient administered concurrent bortezomib), and one G4 infection was reported in a patient administered concurrent chemotherapy; all these severe toxicities were reversible. Pain ‘flare’ during RT was reported only for 8.1% of lesions (47/577) and steroid use during RT had no significant effect on its incidence (*p* = 0.587).

Concomitant systemic treatment had a significant impact on the toxicity reported during RT administration (data available for 410 accesses, *p* < 0.001), with a lower rate of G2 toxicities for patients receiving RT alone and higher rates of G4 toxicity and lower percentages of patients not reporting toxicity in the concurrent chemotherapy group.

The biological dose also had an impact on toxicity: significantly higher rates of grade 1 toxicity during RT were described for treatment with a BED_10_ > 38 Gy (*p* < 0.001), while a BED_10_ ≤ 38 Gy was significantly associated with the absence of toxicity during RT (*p* < 0.001).

Nonetheless, as shown in Table 2, the safety profile was excellent, with no toxicity developed during RT for 59% of the accesses and a toxicity grade ≥2 reported only for 4.1% of the accesses.

Moreover, no significant difference in toxicity possibly attributable to RT was reported at one (*p* = 0.753) or three months (*p*= 0.677) across the groups receiving no CST, biological agents, or chemotherapy.

Similarly, no statistical difference in toxicity was reported at one (*p* = 0.094) or three (*p* = 0.358) months after the end of RT between the groups treated with different BED_10_.

Radiotherapy was completed for 96.6% of the accesses (397 of 411), while definitive suspension due to toxicity was reported only for five accesses (1.2%). In all the accesses that required temporary suspension due to toxicity (1.2%) or other causes (0.5%), RT was resumed and completed.

The results regarding pain control and radiological control are summarized in Table 3. The overall pain control rate at the end of RT was 87.4%: complete control was obtained for 35.9% of lesions and partial control for 51.5%.

There was a statistically significant correlation between BED_10_ and pain control at the end of RT (*p* < 0.001), with higher rates of pain control for a BED_10_ > 38 Gy and lower rates for a BED_10_ < 15 Gy.

Conversely, CST had no impact on the pain control rate at the end of RT (*p* = 0.244).

The data regarding pain control at 1 month were available for 367 lesions. Overall, the complete pain control rate was 66.5%, and the rate of partial control was 30.5%; only 3% of patients had no pain relief on irradiated sites. Radiotherapy alone was significantly linked with higher rates of complete control and lower rates of partial control, and biological treatment was associated with no pain control (*p* = 0.034). This finding could have been biased by the extremely low rate of absence of pain control. On the other hand, BED_10_ had no significant effect (*p* = 0.112) on the pain control rates (*p* = 0.175).

At 3 months (with pain assessed for 355 lesions), the complete pain control rate increased to 73.5%, and the partial and no-control rates were 23.4% and 3.1%, respectively. Radiotherapy alone was significantly linked with higher rates of complete control, concurrent chemotherapy with partial control, and biological treatment with no control (*p* = 0.001). Again, BED_10_ had no significant effect (*p* = 0.112) on the pain control rates at three months.

At six months (with data available for 352 lesions), pain control was complete for 77% of the lesions and partial for 17%, while for 6% of the lesions, pain was not controlled. No significant differences in pain control at 6 months were reported for the different systemic-treatment (*p* = 0.155) or BED_10_ (*p* = 0.499) groups.

The data regarding the radiological response at 6 months after RT were available for 181 lesions. Overall, local complete response (CR) was observed for 8.8% of the lesions, partial response (PR) for 70.2%, stable disease (SD) for 16.6%, and progressive disease (PD) only for eight lesions (4.4%).

Progressive disease was significantly more frequent for the lesions treated without CST, while the administration of concurrent biologic therapy resulted in significantly lower rates of PD (*p* = 0.044). Due to the multitude of different CSTs, the numbers were not sufficient to allow conclusions regarding the impact of each single drug on tumor response.

The impact of BED_10_ on the radiological response at 6 months was less clear, and it was possibly biased by the small number of some samples (e.g., data available for only 11 lesions treated with a BED_10_ < 15 Gy). A BED_10_ > 38 Gy significantly correlated (*p* = 0.001) with lower rates of SD and CR and higher rates of PR, while a BED_10_ of 15–38 Gy resulted in higher rates of SD and CR and a BED_10_ <15 Gy resulted in lower rates of PR and higher rates of PD.

## 4. Discussion

In contrast with solitary plasmocytoma, for which RT is prescribed with curative intent [4,12], radiation treatment for MM is prescribed with palliative intent [4].

Radiotherapy exerts its effect through multiple mechanisms, including the apoptosis of cancer cells, the decompression of nervous structures, the reduction in tumor-associated inflammatory molecules, the inhibition of osteoclasts, and the remineralization of bone [13].

Despite the widespread use of RT for the symptomatic treatment of MM, multiple issues still have to be clarified, including the impact of the RT dose and the possible effect of CST.

### 4.1. Radiotherapy Dose

The definition of the best dose and fractionation of RT for MM is debated, as the current literature is mostly composed of limited retrospective series.

While initial analyses revealed an extremely high rate of symptom relief without a dose–response relationship [14], subsequent reports generally described higher pain control and recalcification rates for higher doses [15,16].

Only a few studies normalized the dose as BED. Among these, in a series of 153 patients (based on available information from 81 patients for pain relief and 69 patients for recalcification), the equivalent dose in 2-Gy fractions was significantly associated with increased rates of pain relief and recalcification [13]. Conversely, no impact of BED was reported in another cohort of 149 patients [2].

Another paper, published in 2019, reported no significant differences in pain response in 130 patients treated in the ‘biological era’, but patients who received a dose of 20–30 Gy had a significantly lower risk of pain recurrence compared with those who received lower doses [17].

Similarly, a comparison of short-course (8 Gy single fraction or 20 Gy/5 fr) and longer-course (30–40 Gy in 10–20 fractions) RT for 172 patients with spinal-cord compression from MM revealed better results in terms of motor function from the long-course treatment [18].

Only one randomized clinical trial compared different schedules for the treatment of MM (8 Gy single fraction versus 30 Gy in 10 fr): although the pain response and recalcification rates were similar among the two regimens, the patients in the fractionated treatment group reported better quality of life [19].

The results of these experiences influenced the current ILROG recommendations of 20–30 Gy administered in 5–15 daily fractions for bony sites, with 8 Gy in a single fraction preferred for patients with a dismal prognosis [4].

A study of the largest cohort of MM patients undergoing RT was recently published by Elhammali et al. [20]: 772 patients treated on 1513 sites were analyzed. The majority of the patients was treated with five of more fractions for a total dose of 20–25 Gy. The authors suggested that such doses were sufficient to achieve a low rate of re-irradiation (2.6%) and could provide durable pain relief and high radiological local control. Nonetheless, a BED_10_ ≤ 28 Gy was associated with a significantly higher risk of re-irradiation, and only 82 treated lesions were assessable for radiological response. Unfortunately, no data were provided regarding CST and toxicity.

Our data seem to confirm previous reports, as the RT allowed high rates of pain control at the end of the course (87.4%), which further increased at three months (96.9%) and were substantially maintained at 6 months (94%).

To the best of our knowledge, this is the largest cohort assessing radiological response, with promising results. Disease control was obtained for the vast majority of the lesions, with an overall response rate of 86.8%, and only 4.4% of the lesions developed in-field progression at 6 months.

Consistently with some of the published series, the impact of BED_10_ on the radiological response was somehow unclear and possibly biased by the reduced numbers of some groups. With these limits, a BED_10_ < 15 Gy resulted in significantly higher rates of PD.

On the other hand, although higher rates of pain control were achieved by the last day of RT with a BED_10_ > 38 Gy and lower rates with a BED_10_ < 15 Gy, BED_10_ had no impact on pain control at 3 or 6 months.

Indeed, the assessment of pain response after RT schedules with a low number of fractions (1–5) is biased by the fact that a few weeks might be required to achieve maximal pain relief after treatment.

Thus, as suggested by current guidelines, treatment with schedules characterized by a low BED_10_ (such as 8 Gy in a single fraction) could be an effective option for pain relief, while a BED_10_ > 15 Gy could be a better option if the aim is to achieve local disease control.

The adoption of schedules with higher BED_10_ should not raise concerns of increased toxicity as, although a BED_10_ > 38 Gy resulted in higher rates of side effects, this was limited to reversible grade 1 events, and no difference in toxicity was reported at three or six months after RT between the groups treated with different BED_10_.

### 4.2. Concurrent Systemic Treatment

Even fewer data, all from retrospective studies, are available regarding the potential synergy, interactions, and toxicities of RT and CST.

Although some experiences suggest higher rates of recalcification for patients undergoing concurrent chemotherapy [16], other analyses reported no influence on recalcification [14] or pain response [13]. Moreover, in the majority of the available studies, the characteristics of systemic treatment and timing relative to RT were not clearly specified, and the definition of CST was variable.

Only two studies analyzed the combination of biological agents and RT, with no details regarding the effect on tumor response.

The data from a cohort of 39 patients (64 sites) who received RT with CST for osteolytic lesions were published in 2014 by Shin et al. [10]. RT was completed by 89.7% of the patients and by 14 of 16 the administered novel agents, the toxicity was fair and mainly hematologic, and the clinical pain response was optimal regardless of CST.

Salgado et al. [9] evaluated 130 patients (279 treated sites) who received RT from 2007 to 2017, of whom 91 were administered concurrent (defined as administration from one month before to one month after RT) biological agents, mainly bortezomib, carfilzomib, and daratumumab. No significant difference in the incidence of acute or sub-acute toxicity was reported between the patients who received concurrent biological agents and those who received RT alone, and all the reported toxicities were grade < 3.

Talamo et al. [2] retrospectively evaluated a cohort of 149 patients treated on 262 sites during the ‘biological era’. Unfortunately, the timing of the RT and systemic treatments and possible interactions were not reported, but the RT did not decrease the number of peripheral blood stem cells collected for autologous stem-cell transplant.

In the absence of solid clinical data, the main concern over administering RT concurrently with systemic treatment is the possible sensitization of normal tissues, leading to increased toxicity. Moreover, two case reports suggest the potential of augmented intestinal toxicity when RT is performed on abdominal or pelvic sites concomitantly with bortezomib [11,21].

This could be explained by the radio-sensitizing effect of bortezomib [11] and the rapid turnover of the intestinal mucosa, and other classes of target therapy, such as cyclin-dependent kinase inhibitors [22], were also linked with unexpected severe radiation-induced enteritis [23]. Nonetheless, no increased toxicity was reported for the patients undergoing regimens including bortezomib during RT in our cohort.

To the best of our knowledge, this is the largest series assessing the impact of systemic treatment during RT for MM and the only one to accurately define ‘concurrent’ treatment in the ‘biological era’. The results of our study seem to confirm the safety of RT administered to patients undergoing CST, considering both chemotherapy and target drugs. Although lower rates of G2 toxicities were reported for RT without CST and higher rates of G4 toxicities for concurrent chemotherapy, the safety was excellent overall, and no difference in toxicity was reported at one or three months across the groups receiving no CST, biologic agents, or chemotherapy.

Systemic treatment had no impact on pain control at the end of RT. Surprisingly, the patients who underwent RT without CST reported a significantly higher rate of complete pain control at 1 and 3 months after the end of RT. This finding is likely biased by the extremely low rate of absence of pain control, and it should also be considered that pain control at six months was not affected by the CST.

On the other hand, the radiological disease progression at six months was significantly more frequent for the lesions treated with RT without CST and less common in cases of the administration of concurrent biologic therapy (*p* = 0.044), suggesting a potential synergy with ionizing radiation.

Nonetheless, systemic treatment in the intervening period between RT and radiological assessment could also have had an impact on the response at imaging, which would have been difficult to quantify; this may also have been due to the large numbers, combinations, and sequences of different systemic therapies.

Despite the reassuring reported toxicity profile, when RT is performed on abdominal sites concurrently with potentially radio-sensitizing systemic agents, as a precaution, the treatment plan should be optimized to limit doses to the intestine. In selected cases, conformal treatments, such as intensity-modulated radiation therapy, could aid in reducing doses to the organs at risk, as has already been demonstrated for other hematological malignancies, such as lymphoma [24,25]. If concerns over toxicity persist, relatively low prescription doses could still allow satisfactory disease control [6].

The limits of our analysis must also be acknowledged. The evaluation of pain and its response to treatment is complex, as defining the precise site of pain in relation to RT is not always straightforward for a patient, and several confounding factors (including the use of analgesics and steroids) affect pain control. Although consisting of a large sample, our cohort was retrospective, which could explain the limited proportion of patients for whom complete data regarding the radiological response and the definition of the pain response according to VNS and analgesic drug intake were available. Moreover, some groups (considering both BED_10_ and the type of systemic treatment) were under-represented, limiting the statistical analysis and the potential to define the impact of each single drug.

On the other hand, it should be considered that the main outcomes of this analysis were to assess the tolerability of RT alone or in combination with systemic agents, and the rates of radiological and pain control regardless of BED_10_.

Prospective data from large populations are awaited to confirm our results.

The management of MM is still evolving, and the possible impact of new imaging modalities [26,27] and combinations of treatments with new therapies, such as chimeric antigen receptor T-cell therapies [28] and checkpoint inhibitors [29] could open new perspectives.

## 5. Conclusions

Despite a slight increase in the toxicity rates in the case of CST and the increased BED_10_, the toxicity profile of RT for the treatment of MM was excellent. Irradiation resulted in high pain control rates at the end of treatment, which further improved at three months and were substantially maintained at six months. The radiological disease control was also optimal, with only a few lesions presenting in-field progression. Schedules with a BED_10_ < 15 Gy allowed satisfactory pain control, but resulted in significantly higher rates of radiologic PD, and higher doses are thus suggested if the aim is not purely analgesic. A potential synergy of ionizing radiation and CST was suggested by the significantly higher rates of radiological PD for the lesions treated with RT without CST, and by the lower rates in case of the administration of concurrent biological therapy.

## Figures and Tables

**Table 1 cancers-14-02273-t001:** Patients’ and treatment characteristics. BED_10_ = biologically effective dose (assuming an α/β ratio of 10); 3DCRT = 3-D conformal radiotherapy; 2DRT = 2-D radiotherapy; IMRT = intensity-modulated radiotherapy; VMAT = volumetric modulated arc therapy; Tomo = tomotherapy.

Patients’ Characteristics (Per Access)			
Lines of Systemic Treatment before (Number and Percentage, Data Available for 409 Accesses)			
Mean 1.65		Range 0–10	
One line 42 (10.3%)			
Two lines 65 (15.9%)			
Three lines 38 (9.3%)			
Four lines 28 (6.8%)			
Five or more lines 43 (10.5%)			
**Number of Treated Lesions**			
Mean 1.41		Range 1–4	
One lesion 274 (66.7%)			
Two lesions 110 (26.8%)			
Three or more 27 (6.6%)			
**Age at Start of Radiotherapy**			
Median 68.1 years		Range 31.2–92.5 years	
**Treatment Characteristics (per Lesion)**			
**Systemic Treatment Concurrent with Radiotherapy**			
No 220 (38.1%)	Chemotherapy 76 (13.2%)		Biologic treatment 281 (48.7%)
**Type of concurrent biological systemic treatment**			
Lenalidomide or lenamide-containing combinations 76 (13.2%)			
Bortezomib 67 (11.6%)			
Bortezomib-containing combinations 117 (20.3%)			
Other biological agents 21 (3.6%)			
**Site of treated lesions**			
Vertebral 346 (60%)	Extremities 108 (18.7%)		Pelvic bones 57 (9.9%)
Ribs/sternum 36 (6.4%)	Skull 15 (2.6%)		Other 15 (2.6%)
**Radiotherapy BED_10_**			
Less than 15 Gy 42 (7.3%)	15–38 Gy 165 (28.6%)		More than 38 Gy 370 (64.1%)
**Radiotherapy schedules**			
30 Gy/10 fr 360 (62.4%)	20 Gy/5 fr 158 (27.4%)		8 Gy/1 fr 34 (5.9%)
**Radiotherapy technique**			
3DCRT 440 (76.3%)	2DRT 117 (20.3%)		IMRT/VMAT/Tomo 20 (3.5%)

**Table 2 cancers-14-02273-t002:** Higher-grade toxicity reported during radiotherapy, at one month and at three months after the end of radiotherapy. RT = radiotherapy; CHT = chemotherapy; BED10 = biologically effective dose (assuming an α/β ratio of 10).

Higher-Grade Toxicity (Per *n* of Accesses)						
During Radiotherapy (Data Available for 410 Accesses)						
	G0	G1	G2	G3	G4	Total
Overall	242 (59%)	151 (36.8%)	13 (3.2%)	3 (0.7%)	1 (0.2%)	410
RT alone	101 (64.3%)	54 (34.4%)	1 (0.6%)	1 (0.6%)	0 (0%)	157
Concurrent CHT	22 (43.1%)	24 (47.1%)	3 (5.9%)	1 (2%)	1 (2%)	51
Concurrent biological agent	119 (58.9%)	73 (36.1%)	9 (4.5%)	1 (0.5%)	0 (0%)	202
BED_10_ < 15 Gy	28 (90.3%)	2 (6.5%)	0 (0%)	1 (3.2%)	0 (0%)	31
BED_10_ 15–38 Gy	86 (71.7%)	30 (25%)	3 (2.5%)	0 (0%)	1 (0.8%)	120
BED_10_ > 38 Gy	128 (50.6%)	119 (45.9%)	10 (3.9%)	2 (0.8%)	0 (0%)	259
**At 1 Month after Radiotherapy (Data Available for 267 Accesses)**						
	G0	G1	G2	G3	G4	Total
Overall	251 (94%)	14 (5.2%)	2 (0.7%)	0 (0%)	0 (0%)	267
RT alone	96 (92.3%)	7 (6.7%)	1 (1%)	0 (0%)	0 (0%)	104
Concurrent CHT	21 (100%)	0 (0%)	0 (0%)	0 (0%)	0 (0%)	21
Concurrent biological agent	134 (94.4%)	7 (4.9%)	1 (0.7%)	0 (0%)	0 (0%)	142
BED_10_ < 15 Gy	20 (100%)	0 (0%)	0 (0%)	0 (0%)	0 (0%)	20
BED_10_ 15–38 Gy	67 (98.5%)	0 (0%)	1 (1.5%)	0 (0%)	0 (0%)	68
BED_10_ > 38 Gy	164 (91.6%)	14 (7.8%)	1 (0.6%)	0 (0%)	0 (0%)	179
**At 3 Months after Radiotherapy (Data Available for 263 Accesses)**						
	G0	G1	G2	G3	G4	Total
Overall	254 (96.6%)	8 (3%)	1 (0.4%)	0 (0%)	0 (0%)	263
RT alone	99 (96.1%)	3 (2.9%)	1 (1%)	0 (0%)	0 (0%)	103
Concurrent CHT	20 (100%)	0 (0%)	0 (0%)	0 (0%)	0 (0%)	20
Concurrent biological agent	135 (96.4%)	5 (3.6%)	0 (0%)	0 (0%)	0 (0%)	140
BED_10_ < 15 Gy	18 (100%)	0 (0%)	0 (0%)	0 (0%)	0 (0%)	18
BED_10_ 15–38 Gy	66 (100%)	0 (0%)	0 (0%)	0 (0%)	0 (0%)	66
BED_10_ > 38 Gy	160 (94.7%)	8 (4.7%)	1 (0.6%)	0 (0%)	0 (0%)	169

**Table 3 cancers-14-02273-t003:** Data regarding pain control and radiologic response. RT = radiotherapy; CHT = chemotherapy; BED_10_ = biologically effective dose (assuming an α/β ratio of 10).

Pain Control (per *n* of Lesions)											
At End of Radiotherapy											
		Complete control		Partial control			No control		Total		
Overall		193 (35.9%)		277 (51.5%)			68 (12.6%)		538		
RT alone		70 (34%)		102 (49.5%)			34 (16.5%)		206		
Concurrent CHT		29 (39.7%)		35 (47.9%)			9 (12.3%)		73		
Concurrent biological agent		94 (36.3%)		140 (54.1%)			25 (9.7%)		259		
BED_10_ < 15 Gy		11 (39.3%)		6 (21.4%)			11 (39.3%)		28		
BED_10_ 15–38 Gy		51 (34.5%)		71 (48%)			26 (17.6%)		148		
BED_10_ > 38 Gy		131 (36.2%)		200 (55.2%)			31 (8.6%)		362		
**At 1 Month after Radiotherapy**											
		Complete control			Partial control		No control			Total	
Overall		244 (66.5%)			112 (30.5%)		11 (3%)			367	
RT alone		105 (75.5%)			31 (22.3%)		3 (2.2%)			139	
Concurrent CHT		17 (56.7%)			13 (43.3%)		0 (0%)			30	
Concurrent biological agent		122 (61.6%)			68 (34.3%)		8 (4%)			198	
BED_10_ < 15 Gy		14 (56%)			11 (44%)		0 (0%)			25	
BED_10_ 15–38 Gy		56 (60.9%)			34 (37%)		2 (2.2%)			92	
BED_10_ > 38 Gy		174 (69.6%)			67 (26.8%)		9 (3.6%)			250	
**At 3 Months after Radiotherapy**											
		Complete control		Partial control			No control		Total		
Overall		261 (73.5%)		83 (23.4%)			11 (3.1%)		355		
RT alone		108 (80.6%)		25 (18.7%)			1 (0.7%)		134		
Concurrent CHT		17 (58.6%)		12 (41.4%)			0 (0%)		29		
Concurrent biological agent		136 (70.8%)		46 (24%)			10 (5.2%)		192		
BED_10_ < 15 Gy		14 (66.7%)		7 (33.3%)			0		21		
BED_10_ 15–38 Gy		58 (64.4%)		28 (31.1%)			4 (4.4%)		90		
BED_10_ > 38 Gy		189 (77.5%)		48 (19.7%)			7 (2.9%)		244		
**At 6 Months after Radiotherapy**											
		Complete control		Partial control			No control		Total		
Overall		271 (77%)		60 (17%)			21 (6%)		352		
RT alone		109 (79.6%)		20 (14.6%)			8 (5.8%)		137		
Concurrent CHT		19 (67.9%)		9 (32.1%)			0 (0%)		28		
Concurrent biological agent		143 (76.5%)		31 (16.6%)			13 (7%)		187		
BED_10_ < 15 Gy		15 (71.4%)		5 (23.8%)			1 (4.7%)		21		
BED_10_ 15–38 Gy		65 (73.9%)		19 (21.6%)			4 (4.5%)		88		
BED_10_ > 38 Gy		192 (78.7%)		36 (14.8%)			16 (6.6%)		244		
**Radiological Response at 6 Months (*n* of Lesions)**											
	Complete response		Partial response			Stable disease		Disease progression			Total
Overall	16 (8.8%)		127 (70.2%)			30 (16.6%)		8 (4.4%)			181
RT alone	8 (11.8%)		43 (63.2%)			10 (14.7%)		7 (10.3%)			68
Concurrent CHT	0 (0%)		9 (69.2%)			4 (30.8%)		0 (0%)			13
Concurrent biological agent	8 (8%)		75 (75%)			16 (16%)		1 (1%)			100
BED_10_ < 15 Gy	2 (18.2%)		3 (27.3%)			4 (36.4%)		2 (18.2%)			11
BED_10_ 15–38 Gy	8 (25.8%)		13 (41.9%)			9 (29%)		1 (3.2%)			31
BED_10_ > 38 Gy	6 (4.3%)		111 (79.9%)			17 (12.2%)		5 (3.6%)			139

## Data Availability

The data and material are stored according to our institutional protocols and are available upon request.

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
