# Peer review of "RR Myelo POINT: A Retrospective Single-Center Study Assessing the Role of Radiotherapy in the Management of Multiple Myeloma and Possible Interactions with Concurrent Systemic Treatment"

_cancers, 2022, doi:10.3390/cancers14092273_

Round 1
Reviewer 1 Report
Please see the file in the attachment

Author Response
We would like to thank the reviewer for the insightful suggestions, that allowed us to improve the quality of our paper. Replies are included in the attached file.

Reviewer 2 Report
Review: RR Myelo POINT: a retrospective single center study assessing 2 the Role of Radiotherapy in the management of Multiple Mye-3 loma and POssible INTeractions with concurrent systemic 4 treatment.
The study provides very interesting and important data on the efficacy of radiotherapy in clinically symptomatic lesions of plasmocytoma. It is a pleasure to review this manuscript.
However, there are several minor points, that should be addressed by th authors, to further improve the quality of the manuscript.
First, please introduce abbreviations at the first appearance of the respective word– and then stick to them / use them throughout the entire manuscript!
For example line 20 / abstract:
“ although chemotherapy, biologic agents and radiotherapy…” -> “although chemotherapy, biologic agents and radiotherapy (RT)…”.
In line 26 and 29: do not use “radiotherapy”, use “RT”…
Second: line 26: please introduce “BED”, before using the abbreviation
Further points:
Line 23 – 25:
“ we retrospectively analyzed records of patients that RT for MM at our Institution from January 1, 2005 to June 30, 2020; data of 312 patients and 577 lesions (treated in 411 accesses) were retrieved.”
This sentence seems as if two sentences were unhappily married to each other. Please revise.
Line 25:
“ most of treated lesions involved vertebrae (60%) or extremities 25 (18.9%).”
Please change to: “Most treated lesions involved vertebrae (60%) or extremities 25 (18.9%).”
Line 32 ff:
“ Radiological response rate at six months (data available for 31 181 lesions) was 79% with only 4.4% of lesions in progression, that was significantly…”
Please formulate the sentences so concisely that the reader still has a chance to keep track...
e.g.: “ Radiological response rate at six months (data available for 181 lesions) was 79% with only 4.4% of lesions in progression. Progression was associated with…”
Background:
Line 40:
“Radiotherapy” -> introduce the abbreviation “RT” in the main text, then use it consequently.
Line 49:
“ionizing radiations“ -> please use the abbreviation “RT”
Line 67:
Use the introduced abbreviation for “concurrent systemic therapy”
Methods:
Line 84: Did you also examine the fracture risk of the lesions before and after RT? Was a particular score used for this?
Line 86:
Alternative wording suggestion: “This protocol has been developed, and the study was conducted according to the criteria…”
Table 1: In my view, this table is very confusing. Even though this makes the table appear much more extensive, it would be helpful to use 2 to 3 columns with a corresponding structure. In addition, in my view, the medians could well be omitted.
Line 111 and following:
Please do not use “p 0.000”, instead “p<0.001”, please do not write “p 0.060”, instead use “p= 0.06”
Line 122:
Replace “significative”, use “significant”
Line 124, line 140
Abbreviation for “concomitant systemic therapy”?
Throughout the result and discussion sections:
Use abbreviations like RT etc.
Line 167:
“Radiotherapy alone was significantly linked with higher rates of complete control, concurrent 168 chemotherapy with partial control and biologic treatment with no control (p 0.001).”
- What does this mean? RT alone is more efficient than combined therapy? Is this result “just chance”? You should comment this finding in the discussion.
Discussion
The content of the discussion is excellent!
Institutional Review Board Statement: please provide number and date
Author Response
The study provides very interesting and important data on the efficacy of radiotherapy in clinically symptomatic lesions of plasmocytoma. It is a pleasure to review this manuscript.
However, there are several minor points, that should be addressed by th authors, to further improve the quality of the manuscript.
Firstly, we would like to thank the Reviewer for the recognition of our work and for the insightful hints, that allowed us to improve the quality of the paper.
First, please introduce abbreviations at the first appearance of the respective word– and then stick to them / use them throughout the entire manuscript!
We apologize for the multiple discrepancies with the use of abbreviations, that have been fixed as suggested in the abstract and in the main text.
Further points:
Line 23 – 25:
“ we retrospectively analyzed records of patients that RT for MM at our Institution from January 1, 2005 to June 30, 2020; data of 312 patients and 577 lesions (treated in 411 accesses) were retrieved.”
This sentence seems as if two sentences were unhappily married to each other. Please revise.
The sentences have been reformulated.
Line 25:
“ most of treated lesions involved vertebrae (60%) or extremities 25 (18.9%).”
Please change to: “Most treated lesions involved vertebrae (60%) or extremities 25 (18.9%).”
The error has been fixed.
Line 32 ff:
“ Radiological response rate at six months (data available for 31 181 lesions) was 79% with only 4.4% of lesions in progression, that was significantly…”
Please formulate the sentences so concisely that the reader still has a chance to keep track...
e.g.: “ Radiological response rate at six months (data available for 181 lesions) was 79% with only 4.4% of lesions in progression. Progression was associated with…”
The sentence has been modified as correctly suggested for the sake of readibility
Background:
Line 40:
“Radiotherapy” -> introduce the abbreviation “RT” in the main text, then use it consequently.
Line 49:
“ionizing radiations“ -> please use the abbreviation “RT”
Line 67:
Use the introduced abbreviation for “concurrent systemic therapy”
The adoption of abbreviation has been consistently applied throughout the text
Methods:
Line 84: Did you also examine the fracture risk of the lesions before and after RT? Was a particular score used for this?
Unfortunately, we did not consistently examine the fracture risk with a specific score.
Line 86:
Alternative wording suggestion: “This protocol has been developed, and the study was conducted according to the criteria…”
Table 1: In my view, this table is very confusing. Even though this makes the table appear much more extensive, it would be helpful to use 2 to 3 columns with a corresponding structure. In addition, in my view, the medians could well be omitted.
The Table has been modified for sake of clarity, as correctly suggested.
Line 111 and following:
Please do not use “p 0.000”, instead “p<0.001”, please do not write “p 0.060”, instead use “p= 0.06”
Line 122:
Replace “significative”, use “significant”
Line 124, line 140
Abbreviation for “concomitant systemic therapy”?
These appropriate corrections have been performed.
Line 167:
“Radiotherapy alone was significantly linked with higher rates of complete control, concurrent 168 chemotherapy with partial control and biologic treatment with no control (p 0.001).”
What does this mean? RT alone is more efficient than combined therapy? Is this result “just chance”? You should comment this finding in the discussion.
This finding, although surprising and counterintuitive, is likely due by chance. This has been better specified in the discussion section.
'Surprisingly, patients that underwent RT without CST reported a significantly higher rate of complete pain control at 1 and 3 months after RT end. This finding is likely biased by the extremely low rate of absence of pain control and it should as well be considered that pain control at six months was not affected by CST.'
Discussion
The content of the discussion is excellent!
We would like to deeply thank the Reviewer for the appreciation of our work.
Institutional Review Board Statement: please provide number and date.
Institutional Review Board number and date have been provided.